# Band-Aid on a Bullet Wound—Canada's Open Work Permit for Vulnerable Workers Policy

Eugénie Depatie-Pelletier [1] , Hannah Deegan [1,*] and Katherine Berze [2]

1   Association for the Rights of Household and Farm Workers (ADDPD-ARHW),
    Montreal, QC H2J 1M3, Canada; eugenie.depatie-pelletier@arhw-addpd.org
2   Faculty of Law, University of Ottawa, Ottawa, ON K1N 6N5, Canada; kberz051@uottawa.ca
*   Correspondence: hannah.deegan@arhw-addpd.org

**Abstract:** In June 2019, the Government of Canada implemented the Open work permit for vulnerable workers (OWP-V) policy, authorizing immigration officers to issue open work permits to migrant workers on employer-specific work permits if they demonstrate reasonable grounds to believe that they are experiencing abuse or are at risk of abuse in their workplace. Drawing on research conducted by a community organization on the impact of the policy, this article examines the policy's potential to remedy the problematic effects of the employer-specific work permit and whether it has been implemented efficiently. Semi-structured interviews were conducted with organizations that provide direct legal and social support to migrant workers in Canada. Additionally, two datasets regarding the role of the OWP-V policy in IRCC's employer compliance regime were analyzed. The research concludes that the OWP-V policy cannot be expected to counteract the high risk of abuse imposed on workers through the employer-specific work permit. Numerous barriers were identified that make it difficult for migrant workers to apply for the permit. The small number of OWP-V permits issued in proportion to the number of employers authorized to hire migrant workers makes it unlikely that the policy will significantly impact employers' propensity to comply with the program conditions.

**Keywords:** migrant workers; labour mobility; vulnerability; employer-specific work permit

## 1. Introduction

The number of migrant workers[1] admitted into Canada has expanded rapidly in recent years, with many sectors relying on them as a source of permanent labour supply (Lu 2020; Zhang et al. 2021; Foster 2012, p. 19; Falconer 2020). Workers coming through the "low-wage" stream are issued work permits that are systematically linked to a specific employer and occupation and require employer endorsement to be renewed. These work permits make it difficult to change jobs, producing a power imbalance that favours employers and results in workers having to remain in abuse situations (Regulations Amending the Immigration and Refugee Protection Regulations, SOR/2019-148). In 2019, in an attempt to temper the adverse effects of the employer-specific work permit, the federal government implemented the *Open work permit for vulnerable workers* policy (OWP-V), which provides immigration officers with the authority to issue an open work permit to a migrant worker with an employer-specific work permit who can demonstrate that they are experiencing abuse or are at risk of abuse by their employer.

The purpose of the OWP-V policy is to mitigate the conditions created by the employer-specific work permit. Employer-specific work permits have the effect of restricting workers' ability to freely change employers by requiring workers to obtain a new work permit for

---

1   The term "migrant worker" refers to workers in Canada without permanent resident status, including undocumented workers. Because our research focuses on workers who are issued employer-specific work permits, the term used in this article has a more limited meaning. It refers to workers entering Canada through the Temporary Foreign Worker Program or International Mobility Program.

each new employer and prohibiting them from earning a livelihood while they wait for the new work permit to be delivered. The procedure is lengthy and fraught with uncertainty. Workers who quit or are fired may fail to find another employer–sponsor to access a new work permit in time (before the expiration of their resident status). Additionally, there is no guarantee that such an employer will be granted by the government the validation necessary to sponsor and employ the worker. Even when successful, it takes many months to complete, during which time workers have no choice but to work without authorization to support themselves while they wait for the new permit to be issued, which exposes them to further risks of abuse and labour trafficking (Beatson et al. 2017, p. 154). In this context, to avoid jeopardizing their precarious right to work in the country, migrant workers systematically endure abuse and refrain from quitting or taking any step that could result in being fired—such as reporting abusive employers (Hastie 2017, pp. 33–34) or participating in state inspections of their employers' practices (Regulations Amending the Immigration and Refugee Protection Regulations, SOR/2019-148). The OWP-V policy is supposed to address these problems by allowing workers who are experiencing abuse to maintain their right to earn a livelihood through the streamlined and rapid issuance (five business days for processing) of work authorization for any employer (subject to the general restrictions imposed on all work permits—Immigration and Refugee Protection Regulations, s 183).

Abuse, for the policy, is defined in section 196.2 of the Immigration and Refugee Protection Regulations and covers four types of abuse: physical, sexual, psychological, and financial.[2] As such, the term "vulnerable" in the context of the OWP-V policy refers to migrant workers who have experienced or are at risk of experiencing at least one of the four types of abuse listed in section 196.2. This is a misnomer to a certain extent. It implies that the vulnerability of the workers eligible for the permit arises from an experience with a specific employer rather than the employer-tying measures that place these workers at a high risk of abuse upon their arrival in Canada. By defining only workers that can provide sufficient evidence of abuse as vulnerable and deserving of protection, the policy disregards and obscures the state-imposed structural vulnerability of all employer-tied migrant workers in Canada.

To be eligible for an OWP-V permit, a worker must hold a valid employer-specific work permit or have applied for a renewal of their employer-specific work permit (implied status). Workers apply by filling out an online application and uploading a letter of explanation detailing the abuse and any supporting evidence. When assessing the application, officers can require that the worker attend an in-person or telephone interview with the immigration officer responsible for their application. The officer must have reasonable grounds to believe that the worker is experiencing or is at risk of experiencing abuse in the context of their employment in Canada.

The OWP-V permit is designed as a transitional measure. It is generally not renewable unless the worker meets the particular criteria established by the operational guidelines (to the best of our knowledge, no renewal has ever been granted). To remain in Canada after its expiration, workers must find an employer with a valid Labour Market Impact Assessment (LMIA) and apply for another employer-specific work permit.

The implementation of the OWP-V policy has not been without issue, and even the department responsible for the policy acknowledges that there is room for improvement, explicitly pointing to concerns over lengthy processing delays and a challenging application process (Mojtehedzadeh 2020). Critics (workers' rights organizations, legal clinics, and unions), on the other hand, have maintained that improving the policy would not be enough to address the dynamics facilitating abuse in the first place—tied-work permit schemes (Migrant Worker Centre & The Law Foundation of British Columbia 2022; Mojtehedzadeh 2020). In this context, a research project was carried out by the Association for the Rights of Household and Farm Workers, a non-profit community organization that

---

2   With the COVID-19 pandemic, the operational guidelines were updated to indicate that the violation of certain health and safety requirements could constitute abuse for the purposes of the policy.

works to advance the rights of migrant household and farm workers through research, education, and legal advocacy. This research project sought to evaluate the policy according to the government's stated objectives and identify areas in the coverage and delivery of the policy that needed to be improved. The research culminated in a report that forms the basis of this paper.

This paper, while reproducing in part the content of the original research report, is supplemented by a literature review of the existing research on the effectiveness of various harm-reduction policies that have been implemented within the context of Canada's temporary labour migration programs. The literature review is followed by a section outlining the research objectives and methodology. The next section presents our key findings: barriers to accessing the program, issues with the application process, policy shortcomings, and the employer compliance aspect of the OWP-V permit. The final section presents recommendations to increase access to the policy and improve its delivery and concludes with a discussion on the limited capacity of individual remedies, such as the OWP-V permit, to meaningfully counteract the high risk of abuse imposed on workers by the legal structures of Canada's temporary foreign worker programs.

## 2. Literature Review

Migrant workers admitted into Canada through the country's temporary labour migration programs are guaranteed the same protections and formal rights as Canadian workers. However, the research and literature have extensively documented how the legal structures of these programs produce conditions of "unfreedom" for workers, undermine their capacity to assert rights, and facilitate the mistreatment and abuse of these workers by employers and recruiters (Hastie 2015; Depatie-Pelletier 2018; Strauss and McGrath 2017; Faraday 2012; Vosko et al. 2019; Beatson et al. 2017; Dumont-Robillard 2019). The federal Standing Committee on Human Resources, Skills and Social Development and the Status of Persons with Disabilities has recognized the negative impact of employer-specific work permits on the physical and mental wellbeing of migrant workers and recommended that the federal government "take immediate steps to eliminate the requirement for an employer-specific work permit" (Canada Parliament 2016, p. 31). The federal government has acknowledged that the employer-specific work permit "can create some conditions under which risk of abuse could be higher" (Regulations Amending the Immigration and Refugee Protection Regulations, SOR/2019-148).

However, Marsden (2019) notes that while policy discourse for the last decade is "replete with the language of "rights" and "protection," for migrant workers", the federal government has failed to undertake reforms that would address the structural conditions that produce migrant workers' increased vulnerability to rights' violations and abuse (p. 155). Instead, the government has preferred an approach that focuses on rights and enforcement (Marsden 2019, p. 161). Enforcement and inspection powers were expanded in 2015, and immigration regulations were amended to require employers to comply with the applicable federal and provincial laws that regulate employment and recruitment, as well as make reasonable efforts to provide a workplace free of abuse (Tucker et al. 2020, p. 8). The 2018 federal budget saw increased funding for compliance and employer inspections (Marsden 2019, pp. 154–55). Measures intended to increase migrant workers' knowledge of their rights in Canada have been undertaken and resources allocated to a network of migrant worker support organizations (Marsden 2019, p. 155). The OWP-V policy was implemented in this context, as part of the "Government of Canada's ongoing commitment to migrant worker protection" (Regulations Amending the Immigration and Refugee Protection Regulations, SOR/2019-148).

The measures implemented by the government do reflect recommendations supported by migrant rights organizations and researchers (Canadian Council for Refugees 2016; Foster and Luciano 2020, pp. 50–52). However, in the absence of reforms to the laws and policies that produce migrant worker "unfreedom", the literature is critical of the capacity of these measures to improve material conditions for workers. An evaluation of the federal

inspection and enforcement system by Tucker et al. (2020) finds that the new system to protect migrant workers from rights violations is inefficient (p. 26). Federal investigators cannot independently determine whether a violation of the applicable protective employment laws has occurred but must rely on a finding of non-compliance by the federal/provincial/territorial authority responsible for the law's enforcement (ibid., p. 14). No matter the non-compliance recognition process, the legal structures that discourage migrant workers from complaining in case of employer abuse have remained largely intact and the federal enforcement system is structurally dependent in part on worker testimony, the researchers conclude that the new system "does not increase the likelihood that violators will be detected" (ibid.). Regarding the measures designed to increase migrant workers' knowledge of their legal rights and support them in the exercise of these rights, Hastie (2017) presents compelling arguments on why such measures are likely to have a limited impact when the structural conditions that prevent migrant workers from exercising or enforcing their rights remain in place. Similarly, Marsden (2019) contends that the approaches focused on legal rights education will be ineffective at delivering large-scale change if the immigration laws and policies that produce migrant workers' vulnerability remain unchallenged (pp. 173–74).

As far as this recent OWP-V policy is concerned, there does not seem to be any academic literature yet on the subject, but two community organizations felt compelled to analyze its impact on the ground (Migrant Worker Centre & The Law Foundation of British Columbia 2022; Association for the Rights of Household and Farm Workers 2021). However, parallels can be drawn between the OWP-V policy and the Temporary Resident Permit for Victims of Human Trafficking (VTIP TRP). Introduced in 2006, the VTIP TRP is supposed to "provide protection to vulnerable foreign nationals who are victims of human trafficking, by regularizing their status in Canada" (Brown 2007). The directives establishing the VTIP TRP allow an immigration officer to issue a short-term temporary residence permit if they determine that the applicant is a victim of trafficking, and the regularization of status is warranted. Studies on this policy have found many issues with the implementation of the VTIP TRP, issues that have the potential to be reproduced in the context of the OWP-V policy. Ricard-Guay and Hanley (2014) report that the application process was considered intrusive, fraught with uncertainty, and difficult to access in practice (pp. 81–83). Other shortcomings include inconsistency in decision-making, significant delays in processing, a discrepancy between how immigration officers and communities define and identify situations of trafficking, as well as a risk of deportation for survivors that come forward but are not issued a VTIP TRP (Canadian Council for Refugees 2013).

## 3. Research Objectives and Methods

The objectives of the research were twofold. The first was to evaluate whether the OWP-V policy has been designed in a manner that supports the government's stated goals, which are as follows (Regulations Amending the Immigration and Refugee Protection Regulations, SOR/2019-148):

1. To provide migrant workers experiencing abuse, or at risk of abuse, with a distinct means to leave their employer (i.e., by opening the possibility of obtaining work authorization for other employers);
2. To mitigate the risk of migrant workers in Canada leaving their jobs and working irregularly as a result of abusive situations;
3. To facilitate the participation of migrant workers experiencing abuse, or at risk of abuse, in any relevant inspection of their former employer and/or recruiter or in otherwise assisting authorities by reducing the perceived threat and the fear of work permit revocation and removal from Canada.

As such, the research sought to assess whether the OWP-V policy has the potential to serve as a remedial policy capable of negating the problematic effects of the employer-specific work permit, in particular the increased risks of abuse and forced labour, the

increased risk of unauthorized employment, and the widespread impunity of employers and recruiters who abuse migrant workers.

The second objective of the research was to evaluate whether, regardless of any limitations in the measure's large-scale impact, the OWP-V policy has been implemented in accordance with the government's acknowledgement that the administrative procedure is intended to respond to the urgent needs of vulnerable individuals.

Two different research methods were used to collect data for the evaluation. The first involved ten semi-structured interviews with organizations and individuals that provide direct legal and social support to migrant workers in Canada. The second method consisted of a direct data request submitted to Immigration, Refugees and Citizenship Canada (IRCC) on OWP-V policy operations within the framework of IRCC's employer compliance regime.

### 3.1. Semi-Structured Interviews

We conducted semi-structured interviews between August 2020 and February 2021 with 10 participants located in Quebec, Ontario, and British Columbia. These participants were representatives of legal clinics, workers' rights organizations, and a union that provide direct support services to migrant workers. At the time of their interviews, the ten participants indicated that, in total, they had assisted or intervened in approximately 192 individual applications for an open work permit, making them well-placed to identify the limits and recurring issues with the policy.[3] For more details about the recruitment of the participants, the conduct of the interviews, and the analysis of the data, please see Appendix A.

A questionnaire for the interviews was developed by reviewing the submissions to IRCC made by various migrant rights advocacy groups in the lead-up to the creation of the OWP-V program to establish which potential barriers and issues with the policy had already been identified. After analyzing the submissions, we formulated questions to generate information on these possible barriers and challenges. The questions were developed to be as open-ended as possible to allow participants to discuss issues with the policy that had not been identified explicitly during the pre-implementation consultation process conducted by IRCC.

We recognize that the number of interviews is limited, and findings are not representative of all provinces in Canada. Still, the interviews helped to identify critical barriers and challenges that workers face when applying for the permit and, as such, contribute a valuable source of information on this new and understudied topic.

When the research was conducted, the physical isolation experienced by migrant workers, which in normal times poses a challenge for research on this population, was further exacerbated due to the COVID-19 pandemic. Furthermore, any research that directly involves migrant workers must be designed and undertaken with particular attention to the specific vulnerabilities of the population. The project did not have the resources necessary to recruit migrant workers and facilitate their participation properly. For all these reasons, the research did not include the direct inputs of migrant workers themselves, although they would have undoubtedly contributed valuable data on many aspects of the evaluation. We also recognize that despite the important insights that individuals and organizations who support migrant workers can provide, those perspectives are not a substitute for the lived experiences of those directly impacted by the OWP-V policy. Thus, future research on the policy that would integrate the perspectives of the migrant workers themselves is urgently needed.

---

[3] During the interviews, it was revealed that there is a considerable collaboration between organizations when assisting with an OWP-V application. It is very possible that multiple participants intervened or assisted with the same application. We did not confirm which applications received assistance from more than one participant/organization. As such, the actual number of workers that had benefited from the help of the participants at the time of our study is less than 192.

*3.2. IRCC Data Request*

The OWP-V policy was implemented to address the negative impact of the employer-specific work permit on migrant workers' (lack of) propensity to report abusive employers and facilitate their participation in inspections and/or investigations of their employer by state authorities. Thus, the policy is supposed to support "the worker protection objectives of existing employer compliance programs" by helping to increase the identification of non-compliant/abusive employers and the imposition of consequences on such employers. In fact, when an OWP-V application is approved, a summary of the allegations (an OWP-V inspection referral) is sent by the processing IRCC officer to the branches responsible for inspections and employer compliance.

To evaluate whether the OWP-V policy did, in fact, lead to better law enforcement against non-compliant/abusive employers, IRCC and Service Canada agreed to produce and share specific datasets regarding the following:

- The number of times an inspection was triggered due to an OWP-V permit being issued.
- How those inspections were conducted (remotely, scheduled, etc.).
- The number of employers that had been found non-compliant due to an inspection that the issuance of an OWP-V permit had triggered.
- The penalties or consequences, if any, that were imposed on those employers. How many OWP-V permits automatically issued to migrant workers associated with an employer found to be non-compliant and with their authorization to employ a temporary foreign worker revoked?

IRCC responded on 23 October 2020 to these questions, which contained two sets of data regarding OWP-V inspection referrals and outcomes. This is because Service Canada inspects employers of workers under the Temporary Foreign Worker Program (TFWP), while IRCC administers those under the International Mobility Program (IMP). The data were analyzed to establish to what extent approved OWP-V applications led to inspections of employers and what resulted from those inspections.

## 4. Findings

*4.1. Barriers to Accessing the Program*

The participants were asked to identify recurring factors that prevented or made it difficult for workers experiencing abuse or at risk of abuse to apply for the OWP-V permit. In the following section, we focus on these factors: employer-dependent legal status, the valid work permit requirement, no funding for legal support and assistance, no independent access to transportation and communication technologies, and the desire to avoid further psychological distress.

### 4.1.1. Employer-Dependent Legal Status

Canada's temporary foreign worker programs make workers' legal status in the country dependent on employers in many ways, and workers that disclose abuse risk facing reprisals from their employers and, more importantly, from state agencies in case of termination of employment (Depatie-Pelletier 2018). The potential negative consequences of coming forward with a complaint, for both a worker's current and future legal status in the country, are indisputably grave. The participants repeatedly flagged workers' legitimate concerns about safeguarding their legal status as undermining the efficacy of the exceptional issuance of an open work permit to address employer-tied worker abuse.

For instance, the participants mentioned that fear of reprisals by employers made workers hesitant to apply for the permit. If the worker is still employed with the employer, they are concerned that the employer will find out about the application. As one participant explained, "they're also of course very fearful of immigration, of their employer finding out that they need an application. I've seen examples of someone making an application that was not successful, and then Service Canada visited their worksite and was asking



questions, and the employer put two and two together and terminated the employee, leaving them in an even worse position than they started."

Workers with a permit near expiration might be concerned that disclosing their situation to immigration authorities may trigger deportation proceedings (before having secured another employer-sponsor) if their application is refused. This concern, said one participant, is "a big psychological burden for the workers." In the case of agricultural workers, the Seasonal Agricultural Worker Program (SAWP) structure requires that workers return to their country of origin every year and may not be allowed back if not re-sponsored by a member of a coalition of agricultural employers. It is well documented that there is "a real risk of not being retained or recalled to work in subsequent seasons, otherwise known as "blacklisting ( . . . )" (Marsden 2019, p. 168). The participants confirmed that migrant farmworkers were often reluctant to apply for the permit because they did not want to jeopardize future renewals of their Canadian work permit by employers. As one participant explained, "workers fear that if they take this step that they will face blacklisting, and it is important to maintain access to the program because it is an important source of income for them and their families."[4] Indeed, one Ontario researcher confirmed, during the period of this evaluation, that SAWP workers employed in the agricultural sectors located in Leamington and Vancouver had been warned by their foreign government representatives to not apply for the open work permit in the case of abuse by their Canadian employer if they wanted to avoid jeopardizing their access to future work permit renewals (Hennebry 2021).

Finally, the participants noted that some workers do not want to apply for the permit because they are worried about what impact the application might have on future permanent immigration procedures. As one participant explained, in the case of migrant caregivers, "it's just that if you say something wrong, it can lead to the wrong result. And they're aware of that . . . They're aware that anything that they say, such as having worked for another employer in the country, could be used against them in the future for their permanent residency."

### 4.1.2. The Valid Work Permit Requirement

In order to be eligible for the OWP-V permit, a worker must hold a valid work permit or have implied status, which means they must have submitted a work permit extension application for the same employer and be awaiting a decision (Immigration and Refugee Protection Regulations, s 207.1). Many of the participants discussed how this requirement prevented some workers who had experienced abuse from benefiting from the policy. One participant stated that "We've had clients whose work permits had already expired by the time that they reached us, and we had to advise them that they were not eligible for this particular program." The participants also indicated that the requirement was a significant problem when workers experienced abuse near the end of their work permit. As one participant shared with us, "She was fired with a work permit that was just going to expire in a few days. She came looking for support, and we were able to help her with her health and well-being because she was distressed. But she wasn't able to get the OWP-V permit even if she had a lot of evidence about the abuse."

One form of abuse often experienced by migrant workers is fraud and deception by employers/recruiters regarding the steps taken to renew their immigration status (Faraday 2014, p. 68). However, those workers cannot access the policy even though the fraud and deception may have been accompanied by other forms of abuse. In these cases, the participants explained that it was difficult to find a solution that would allow the worker to stay in Canada and continue working legally. The participants indicated that they might try to assist workers with applying for a temporary resident permit for victims of human trafficking (VTIP TRP). Still, that approach could not be considered a feasible

---

[4]  The participants shared that workers who are admitted under the Primary Agriculture stream also voiced concerns that applying for the OWP-V permit would mean decreased chances of securing a renewal of their Canadian agricultural work permit in the future, either when their OWP-V permit expired or if they returned to their country of origin and had to be re-admitted once again through the TFWP.



solution in most cases. As one participant explained, "The threshold for getting a resident permit for victims of trafficking is very high, and there's going to be lots of people who experienced abuse, but it may be challenging for the officer to see a link to the definition of human trafficking based on the facts, so I do think that there are definitely people that fall through the cracks."

### 4.1.3. No Funding for Legal Support and Assistance

Throughout the research, it was revealed how much time, support, and assistance goes into a successful OWP-V application. It was apparent that the participants played a vital role in ensuring that workers could navigate the policy and secure the OWP-V permit.

When the participants were asked to describe what services or forms of support they offered to applicants, their answers were generally divided into two categories. Seven of the participants provided help completing the application. In contrast, two of the participants provided psycho-social support to applicants, either during the application process or after the permit had been delivered. One participant's intervention included both direct assistance with completing the application and the provision of psycho-social support to workers.

The participants involved with the administrative procedure indicated that they provided at least one of the following to workers when assisting with an OWP-V application:

- Translation services;
- Plain language explanations for the policy and relevant legislation;
- Assistance with ancillary complaints (to labour boards, police, etc.);
- Referrals to other organizations for support letters, drafting of the affidavit;
- Instructions as to collection and presentation of evidence;
- Interview preparation;
- Assistance with the uploading of documentation and the submission of the application online.

The support and assistance provided reveal how these applications are time, labour, and knowledge intensive. A robust application takes between 15 and 30 h of assistance to be completed. Due to the many barriers identified by the research, most of the participants explicitly stated that they believed their help or support of a worker facing abusive conditions translated into a higher probability of these workers accessing the open work permit.

However, at the time the research was conducted, there was no public funding provided that would directly support the participants' interventions in these applications. A few participants briefly discussed how their organizations had to find creative ways to redirect existing resources towards supporting these applications. In some cases, the participants assisted with these applications on a pro bono or volunteer basis, limiting the amount of assistance that the participants could realistically provide. As one participant explained, "I just don't have the capacity to take on as many as are coming in, and even then, unfortunately, I can only assist the ones that have some sort of infrastructure in place like a support network."

The majority of the participants explicitly stated that they felt it would be very difficult for a worker to successfully apply for the permit with absolutely no assistance. As an illustration of this point, two participants shared how their organization was able to assist a group of workers in successfully re-applying after the applications that the workers had submitted independently were refused. In one of those examples, the participant explained how the workers had initially been rejected because they did not know how to properly frame and present their experiences, an issue the participant's organization was able to rectify on the second application. This lack of "legal know-how" was a recurring theme in many of the interviews and presents a significant barrier, among many others, to the policy.

Importantly, the participants felt that workers do not always have the legal knowledge necessary to identify the way their experiences would make them eligible for the OWP-V permit or to adequately prepare and present their case. Therefore, the participants'

assistance often involved identifying which elements should be included in the affidavit to present the most robust case possible.

A similar issue was that the workers they assisted had no or little awareness of their rights and specifically of their right to apply for an open work permit. As one participant observed, "Often when workers describe their living conditions, it clearly constitutes abuse, however, the baseline of expectations for workers of what to expect in terms of treatment is so low. A lot of times, they don't recognize that the treatment is abusive because it is so normalized". As for the workers' knowledge of the OWP-V permit, many participants indicated that it was the participant who informed the worker about the possibility of applying for an OWP-V permit after being approached by the worker for support with other issues. Multiple participants indicated that when the workers were aware of the policy, they often heard about it through other workers. Other times, the participants stated that the workers may have had limited awareness that the OWP-V permit existed but did not know the application procedure.

Migrant workers that come to Canada to work in low-wage occupations or the agricultural sector may have limited proficiency in English or French. Many participants indicated that language and/or literacy levels made it difficult for workers to apply for the OWP-V permit. Detailed information on the policy is only available in English and French. Many participants indicated that their assistance involved explaining the policy in the language spoken by the worker and then translating the worker's experience into English or French to draft an affidavit and fill out the form. The participants that assisted workers who spoke English or French stated that those workers still required assistance understanding the information provided on IRCC's website about the policy, as well as with drafting the affidavit and filling out the form.

### 4.1.4. No Community Support for Independent Access to Transportation and Communication Technologies

The majority of participants discussed barriers that stemmed from the migrant workers' geographical and social isolation from support services, particularly in the cases of agricultural workers that live on the farm or caregivers that live with their employer. Workers may not receive support from a community organization and thus may be utterly dependent on their employer for access to a computer and transportation, which can be a problem if the worker wants to seek assistance from a government officer. As one participant explained, "They're not going to ask their employer to drive them to a center to do an open work permit application. . . . they often have to make up reasons as to why they have to leave the property".

It was also pointed out that even workers living in urban areas can be isolated, especially if they live with their employer and/or in a more affluent neighbourhood where there is a lack of resources in terms of legal clinics, shelters, food banks, etc. In these cases, it might be difficult for a worker to find, without help from an officially mandated community organization, the support necessary to navigate the OWP-V application process.

More than half of the participants indicated that workers' lack of computer access made applying for the OWP-V difficult. Two participants said that part of their support to workers was access to computers at the community organization's physical office. Another participant described how they would drive out to meet workers and bring their laptops with them. As one participant explained, "Many workers are completely dependent on their employers for access to the internet or even access to a computer. For example, some workers cannot open the application form on their phone because the forms are only available with a PDF reader to open them".

### 4.1.5. Red Tape to Quit an Abusive Condition: Added Psychological Harm

Workers may also refrain from applying because they do not want to identify as a victim. As one participant explained, "To be able to self-identify as somebody who is facing abuse is an incredibly complex and emotionally fraught exercise. I mean, I've witnessed

some workers who know that what is happening to them is wrong, but they don't want to place blame, for example, on the employer, or they don't want to identify or characterize it in that way because of the complex relationship between workers and their employer".

Some participants indicated that a desire to avoid re-traumatization could be a barrier to accessing the policy. Multiple participants pointed out that preparing an application requires the worker to recount painful and distressing experiences or events, which can trigger or exacerbate symptoms of psychological trauma. Thus, many participants emphasized the importance of building trust and adopting a trauma-informed approach when assisting a worker with their application. One participant observed that "It is incredibly traumatizing for clients to go through this process of having to relive, discuss and recount traumas that they've experienced in the process of preparing the application as well as in the interview if one is held." Also discussed was how applying for the permit was at odds with a coping strategy for some workers: "they've been able to cope with their trauma by minimizing it or saying that it's nothing." Another participant expressed concern that the application process was psychologically distressing for workers, stating, "there is definitely a sort of re-victimization."

*4.2. Issues with the Application Process*

In addition to the barriers that made it difficult for workers to apply for the permit, the participants were asked to discuss any issues that arose during the processing of applications that compromised the objectives of the OWP-V policy and the impact those issues had on workers and their experience with the program. In many cases, these issues made it difficult for workers to have their applications approved. In this section, we focus on these factors: a complicated online portal, the excessive evidentiary burden, excessive delays, gaps in officers' knowledge of the law and the conditions faced by migrant workers, problems surrounding the interview process, and no expedited and accessible review of refusals.

4.2.1. Complicated Online Portal

Six participants indicated that the online portal and access to the actual application presented a certain level of technical difficulty. One participant described the online portal as "crude and confusing even for professionals". Another participant noted that even accessing the appropriate form was difficult, "You have to fill out a survey before you even have access to the online portal. The questions that they ask are not easy to go through. For example, there's one question where you have to identify yourself as a worker, and if you don't classify yourself correctly, it takes you to this whole other train of questions and this whole set of different forms". Other participants indicated that workers relied on them to upload supporting documentation or struggled to find someplace where they could scan their documents.

4.2.2. Excessive Evidentiary Burden

Workers must establish reasonable grounds to believe that they are experiencing abuse or are at risk of abuse in the context of their employment in Canada. Many participants indicated that meeting the evidentiary requirement was often difficult for the worker. The requirement prevented some workers from successfully applying and contributed to delays in both submitting the application and its processing. One participant felt that "quite significant proof is often required for the applicant to meet their burden."

Discussions regarding the existence of sufficient evidence (or, more often, non-existence) were featured in most of the interviews. As one participant remarked, workers do not necessarily know "that they should be keeping proof of all of their circumstances as they go along." Similarly, some participants stated that evidence can be difficult to collect when the worker has been terminated and removed from the workplace/housing. Certain types of abuse are less easy to document, such as psychological harassment. As one participant remarked, "If you don't have physical proof of the abuse that you were suffering, how do

you convince the officer that you were at risk?" Another participant described the problem in the following terms: "It is extremely difficult especially when workers are isolated, like live-in caregivers or agricultural workers on small farms with two to three employees. It is difficult to find proof since there are no witnesses to testify about the violence that that worker experienced." Furthermore, a worker's capacity to document and record things is dependent on them having sufficient legal knowledge to recognize, as it is happening, that their rights are being violated. As one participant explained, "If they have basic knowledge of their rights then, they can prepare their proof. For example, in the case of the farmworkers that our organization meets at the airport, in the following year, they were prepared to write down the proof whenever there's a certain violation, and they know to take photos, etc. But in most cases, they don't know to do that kind of preparation."

Many participants discussed how they could use their repeated experiences with the policy to advise workers on how to collect evidence of the abuse. Importantly, the participants were often essential in either connecting workers with organizations that could provide support letters or directing workers towards professionals who could perform psycho-social assessments, which could then be used to support their applications.

Two participants voiced concerns over requests by IRCC for further evidence and documentation. One participant noted that there were times when they received a request for documentation that had already been submitted. It was impossible or difficult to provide and/or needed to be submitted on an extremely short delay, which they felt could dissuade a worker from continuing an application. Another participant related that the processing of applications had been considerably delayed by requests for further documentation, even though extensive evidence had already been submitted in support of the application.

4.2.3. Excessive Delays

The operational guidelines stipulate that OWP-V applications should be processed urgently (within five business days). This reflects that workers applying for the permit are in situations of distress, either because they are victims of abuse in their workplace or have left because of the abuse and cannot legally earn a livelihood until the permit is delivered.

IRCC's one-year update revealed that average processing times were closer to 40 days. Delays were cited as a serious issue by almost every participant. Some participants had assisted with applications with delays of 3 to 4 months. One participant noted that sometimes the OWP-V application was slower than the normal procedure for changing employers, stating that: "This happened in a couple of instances, where we applied for the open work permit for vulnerable workers, and in the waiting months, the person just ended up finding a job offer somewhere else with a Labour Market Impact Assessment and applying for an employer-specific permit. And that actually came through faster than the OWP-V or roughly on the same timeline".

A common theme was that delays have significant negative consequences for the worker. The participants reported that workers exhibited signs of psychological distress, anxiety, and hopelessness in the face of prolonged delays. As one participant noted, "Most of the workers that we assist are no longer employed by their employer, so they're waiting for many, many months and without any source of income."

Some participants pointed out that in the case of workers that are still with their employer, every day they must wait prolongs their exposure to harm. For workers that must leave at the end of the SAWP season, delays are a huge source of stress; as one participant stated: "There is a lot of anxiety because the worker knows that they have given up their re-entrance into the program the following year. Now they are unable to work, and the processing times cause the worker to lose faith, especially when money starts to run out". Long delays also undermine one of the objectives of the OWP-V policy, which is to mitigate the risk that people will engage in unauthorized employment. As one participant pointed out, "You're not achieving that goal when you're taking four months to process an application, people need to work to support themselves."

### 4.2.4. Gaps in Officers' Knowledge of the Law and the Conditions Faced by Migrant Workers

Some participants indicated that their assistance involved ensuring that certain information is included in the application to address the gaps in immigration officers' knowledge of provincial employment, health and safety, and human rights legislation, as well as the conditions of the Temporary Foreign Worker Program. One participant stated that "What we have learned is that we have to name the clause, name the policy to say for instance 'according to the contract, clause 4.3 says this and that,' and we have to attach that to their statement. And also, we have to include in our support letter that this violates conditions under the TFWP or under the human rights code or labour standards."

One participant noted that they had seen immigration officers dismiss evidence that showed the payment of recruitment fees as proof of financial abuse on the basis that the fees were paid "willingly" by the worker, disregarding the fact that charging fees is prohibited by provincial law and that recruitment fees have been recognized as a major factor facilitating the abuse of migrant workers (ITC-ILO 2018, p. 7). This issue is particularly concerning since it means that a worker must not only convince an immigration officer that they did, in fact, experience abusive treatment, but they must also have the legal knowledge necessary to persuade the officer that said treatment is prohibited and would justify the issuance of an OWP-V permit. More simply put, the participants assisted workers not just in establishing the factual grounds that would lead to the issue of an OWP-V permit but also with the legal grounds as well. One participant explained the high rate of approvals for applications they assisted within the following terms: "One of the reasons we have been successful is because we have been bringing the cases to the immigration officers and explaining why it constitutes abuse."

### 4.2.5. Burdensome Interview—Scheduling and Obstacles to Accompaniment

When considering an application, an immigration officer can request either an in-person interview or a telephone interview with the worker. The immigration officer can also waive the interview entirely and elect to do a paper review of the application. Certain participants reported that the scheduling of interviews posed a severe problem for workers. Workers who are still with their employer have very little free time or flexibility with their work schedule, and interviews are scheduled during regular business hours, which means that workers must find a way to take time off to attend the interview and, as a result, risk facing sanctions and/or being fired. This is just not feasible for many workers, as one participant explained, "They have to actually get out of the house, and for some workers, that's not possible when they're actually being required to work the entire day without a break, which is [the case for] many of our clients."

This problem is compounded by the fact that interviews are often scheduled at very short notice and with no consultation with the worker. If the worker is not located close to the interview location, they will have to incur costs to attend the interview. Even when a telephone interview is held rather than an in-person interview, workers might not have the privacy or liberty to participate correctly. As one participant recounted, "This worker when she was called by the officer, she could not leave the farm. [Even] the officer was very concerned that someone would approach and ask 'hey, who are you talking to' and it was very windy, it was challenging."

Many participants also expressed concern that there was no official policy regarding a worker's right to be accompanied during an interview. Information disclosed during the interview can impact whether the OWP-V permit is granted. Accordingly, the participants felt that workers would greatly benefit from having a support person present in the interview as they could help clarify questions. This is especially important given the high risk of confusion due to linguistic barriers, even when translation is provided. Similarly, a couple of participants felt that the interview process could be very intimidating for workers. Workers' vulnerability could make it difficult for them to properly communicate to the immigration officer what they had experienced, to the detriment of their application.

Another participant pointed out that in other contexts, support organizations are generally allowed to accompany the worker and intervene on their behalf: "[At] the labour board, where we have a certain right to be engaged, but it's not the case for the IRCC. So, the official way of intervention and accompanying by our organization is limited."

### 4.2.6. Refusals

The way refusals are decided, and the fact that there is no specific review process by which a worker can automatically challenge a refusal, was cited as highly problematic by the participants.

Concerns were raised about how officers are making their findings and the conclusions they are drawing from the evidence presented. For example, one participant described how they had received the reasons for refusal when they filed for judicial review and discovered that the officer had based their decision on extrinsic evidence to which the worker had not been allowed to respond. Whether immigration officers are deciding applications with due regard to the principles of procedural fairness is beyond the scope of this research. Still, specific comments made by the participants indicate that this may be an important area for further research.

A refusal of an OWP-V application has grave consequences for the worker. If they are still employed with their authorized employer, it means they must remain in a situation where they are experiencing abuse. For a worker that has left their employment or has been fired, a refusal means they cannot legally earn a livelihood while they try to secure another employer-sponsor, which is still required for a work permit renewal. The participants noted that there was no simple and accessible way for a worker to challenge a refusal, even if the refusal was egregious and obviously incorrect.

In these instances, it was explained that workers had three options by which to try and challenge the refusal: to reapply, to submit a request for reconsideration, or to apply for judicial review. However, the participants felt that none of these three possible pathways to challenge an open work permit refusal were genuinely adequate. According to one participant, re-applying offers a low chance of success since "making an unsuccessful application initially seems to be quite fatal, even if additional evidence and submissions are provided to refute some of the adverse findings made in the previous application." On the other hand, judicial review can be a lengthy, costly, and complex process that realistically requires legal representation. Additionally, by the time the application for judicial review is resolved, the worker's work permit may have expired, rendering them ineligible for the OWP-V permit. Finally, as for requests for reconsideration, there is no guarantee that the request will even be granted, and most workers might not even be aware of that option.[5]

### 4.3. *Shortcomings of the OWP-V Program*

While the OWP-V policy provides a mechanism for migrant workers to leave an abusive workplace, the participants noted that there were issues or shortcomings with the program that produced adverse outcomes for workers and were counterproductive considering the purpose of the OWP-V program.

### 4.3.1. Loss of Healthcare and No Accompanying Support Services

One participant who provided psycho-social support to workers pointed out that workers who move from an employer-specific work permit to an OWP-V lose access to public healthcare.[6] The participant felt it was counterintuitive to create a policy that is

---

[5]   A request for reconsideration is an informal process that allows a refused applicant to request, among other things, that the decision maker correct a clerical error or other error, take into consideration new evidence (facts that arose after the original decision was made and communicated to the applicant), or consider additional evidence that was not available at the time of the original decision. The decision to reconsider is discretionary.

[6]   Certain provinces make coverage by provincial insurance plans dependent on the validity of the employer-specific work permit. Depending on the province, workers may lose their access to the provincial health insurance plan once they are issued the open work permit.

supposed to help workers that have experienced abuse but not ensure that those workers would have continued access to healthcare if they are issued the permit. Similarly, multiple participants pointed out that while the OWP-V permit provided workers with greater labour mobility, workers that have experienced abuse need more than just the ability to find another employer. As one participant stated, "It's very problematic that you have to show the government that you are experiencing abuses, all kinds of harms, psychological, physical, then you know it's like ok we give you this band-aid, one-year open work permit and do whatever you need to do over there. It's very, very problematic. It's not a solution. But it's better than nothing if you are experiencing abuse, but it's not much better."

Many of the participants mentioned that workers were often dealing with the loss of housing and were relying on shelters. Similarly, workers on temporary status cannot access social assistance programs, which they often desperately need since a common form of abuse is financial, meaning workers that escape abusive employers often have little or no financial resources to support themselves. As one participant observed, "The policy only takes care of one minimal part of someone's life; it doesn't give them access to anything else. What are you supposed to do when you're looking for an employer, even though you have an open work permit if you have no money, no address, and no place to stay?"

### 4.3.2. Return to Employer-Specific Work Permit: Added Psychological Harm

The participants reported that the workers they assisted were issued OWP-V permits that were valid for 12 months. As the OWP-V permit is for all intents and purposes almost impossible to renew, workers find an employer with a valid LMIA and apply for another employer-specific work permit in order to remain in Canada upon its expiration. Half of the participants pointed out that this requirement caused various problems for workers.

Some participants indicated that 12 months was not long enough to allow the worker to stabilize their situation (emotionally and financially). It was not always possible for the worker to find an employer who would be willing to apply for an LMIA so that the worker could access an employer-specific worker permit before the 12 months was finished. Additionally, even if the worker could find an employer, the return to the employer-specific work permit placed them back in a situation where they were vulnerable to abuse. As one participant commented, "The tied-work permit is the problem because there is a power dynamic between the worker and the employer because the worker cannot move jobs. Employers can basically exploit them, and the worker cannot move. So, what is the logic of having a policy that allows them to leave an exploitative situation but then requires them to return to another one".

This concern that workers are being forced back into a situation where they are vulnerable to abuse is not theoretical, as one participant recounted, "I had one circumstance in which a worker was on an open work permit, and then was recruited onto a closed work permit maybe six months into the open work permit. Then there was abuse on this second closed work permit, so now we have to reapply for the open work permit again". The participants also indicated that workers were still paying recruiters to find employers who could hire them on an employer-specific work permit, meaning that workers are incurring debt to return to the program.

Finally, the requirement that a worker returns to the employer-specific work permit can be distressing for workers dealing with trauma due to the abuse and who realize that they must put themselves back in a position where they will once again be highly vulnerable. One participant, who provided psycho-social support to a worker whose employer had sexually assaulted her, explained, "The idea of going back onto a closed work permit is just horrifying for her because she sees that it's an exploitative system, and she sees . . . the power that the employer has over (the workers) in the system".

### 4.4. *The Employer Compliance Aspect of the OWP-V Policy*

The third objective of the OWP-V policy is to facilitate the participation of the migrant workers in inspections and/or investigations of their employer and, as such, increase the

identification of non-compliant/abusive employers and the imposition of consequences on such employers. The data from IRCC did provide information regarding how many inspections occurred as a result of an OWP-V application and how those inspections were carried out. It also clarified how OWP-V permits are made available to workers employed by employers that are found to be non-compliant. However, it was not possible to use the data provided by IRCC to systematically link inspection outcomes to the allegation/tip that initiated the inspection.

### 4.4.1. Unknown Inspection Outcomes

The Regulatory Impact Analysis Statement published at the launch of the OWP-V program stated that the number of compliance inspections and their outcomes triggered by OWP-V inspection referrals would be tracked as part of the ongoing performance measurement strategies for the TFWP and IMP. However, when the data were requested, Service Canada did not have a tracking system to link inspection outcomes with the specific trigger (such as a complaint, anonymous tip, or OWP-V referral). While the bulk of the OWP-V referrals ends up being handled by Service Canada, the department's failure to systematically track the outcomes of inspections triggered by the OWP-V referrals makes it difficult, using the data provided by IRCC, to assess whether the OWP-V policy has a positive impact on the efficacy of the TFWP's compliance regime.

In terms of the 28 inspections conducted by IRCC due to an OWP-V referral, three employers were found compliant, two were found compliant with justification, and 23 were still being inspected (as of 23 October 2020). In conclusion, at the time of the research, not one employer had been found non-compliant by IRCC based on a complaint of abuse submitted by a migrant worker.

### 4.4.2. Number of Inspections Conducted as a Result of an OWP-V Referral

According to IRCC, an OWP-V inspection referral may not always trigger an inspection since it is possible that the employer in question is already under inspection or that multiple referrals might be made concerning the same employer. As such, there is a discrepancy between the number of OWP-V inspection referrals sent out and the number of inspections conducted. It was reported that between June 2019 and 3 September 2020, IRCC received 94 inspection referrals, launched 28 inspections, had five inspections pending, and was still reviewing and assessing four referrals. Another four of those referrals involved an employer that was already under inspection. The remaining 54 referrals resulted in "other actions", such as being forwarded to Service Canada since the OWP-V permit had been issued to a worker in the TFWP. During roughly the same reporting period, 364 OWP-V inspection referrals were received by Service Canada, and 118 inspections were launched as a result.

### 4.4.3. Conduct of Inspections

According to IRCC, routine inspections of employers are usually "desk-based", which means employers must submit documentary evidence to demonstrate compliance with regulatory requirements. IRCC also indicated that "due to the potentially egregious nature of the allegations which often result in OWP-V referrals," inspections triggered by OWP-V referrals are typically conducted on-site. However, because of the COVID-19 pandemic, all inspections conducted by Service Canada were conducted virtually, starting on 24 April 2020. On-site inspections by Service Canada only resumed on 17 August 2020. Even then, only the most egregious allegations led to on-site inspections and said inspections were scheduled, meaning the employer was given advance notice of the visit.

### 4.4.4. No Automatic OWP-V Permits for Workers Employed by Non-Compliant Employers

It is important to note that when an employer is temporarily banned from hiring a migrant worker, because of non-compliance, all migrant workers working for that employer may see their work permit revoked, leaving them without the authorization to be legally em-

ployed in Canada (Immigration and Refugee Protection Regulations, s 30, McCallum 2016, MI11). The possibility of having one's work permit revoked is a compelling incentive for migrant workers to hide abuse and, thus, shield their abusive employer from the authorities. Furthermore, workers who cooperate with federal (or provincial) inspectors and disclose abuse often experience reprisals from their employers in the form of termination, which significantly discourages other workers from coming forward during inspections. Workers' legitimate concerns about safeguarding their capacity to remain in the country and earn a living legally have repeatedly been flagged by researchers as a factor preventing workers from voicing complaints or speaking-up about violations (Hastie 2017). This undermines the capacity of inspections to uncover abuse in cases where employers use strategies to conceal or hide their non-compliance (Caxaj and Cohen 2019, p. 7; Migrant Workers Alliance for Change 2020, p. 6; European Union Agency for Fundamental Rights 2018, pp. 20–21).

However, according to IRCC, there is no protocol in place that would allow the workers of an employer who is found non-compliant to be issued an OWP-V permit automatically. Those workers would still have to independently apply for an OWP-V permit and have their application assessed according to the program's requirements. However, IRCC did state that a finding of non-compliance would be considered supporting evidence for an OWP-V application and could constitute compelling grounds justifying the issuance of an OWP-V permit.

## 5. Discussion

The adoption of the OWP-V policy reflects the fact that other attempts[7] to address the abuse epidemic within Canada's temporary foreign worker programs have largely failed to overcome the adverse effects of tied-work permit schemes (Marsden 2019, p. 42; Tucker et al. 2020). However, it is unlikely that the OWP-V policy will substantially impact the structural vulnerability of migrant workers to abuse.

As a remedy for individual migrant workers facing abuse in the workplace, the policy suffers from many shortcomings. It also has a limited ability to prevent unauthorized employment and stay.[8] Various barriers make it difficult, if not impossible, for migrant workers to even apply for the permit, and those that do manage to apply must overcome considerable challenges to have their application approved. The data produced by the research helped to identify a set of recommendations to address specific issues in the design and delivery of the policy. Many of these recommendations were explicitly put forward by the participants.

In terms of increasing access to the policy and improving application processing, the following recommendations could help mitigate many of the issues identified by the research:

- Removal of the "valid" work permit requirement;
- Redesign of the online platform to simplify it and make it mobile-friendly;
- Engagement with stakeholders to create a training program for immigration officers on how coercion and abuse occur within the context of temporary foreign worker programs;
- Allocation of dedicated funding to organizations to assist with OWP-V applications;
- Enactment of an official policy regarding the right to be accompanied and assisted during every step of the procedure by legal counsel and a chosen support person;
- Directives to immigration officers instructing them to prioritize paper reviews of applications and to require phone interviews only in the most exceptional cases;
- The provision of interpretation services if an interview is required;

---

[7] Attempts include but are not limited to the following: formalization of migrant workers' legal rights, dissemination of legal rights information, increased employer oversight, increased cooperation with provincial agencies, and state-funded community support initiatives.

[8] Almost half of applications are refused, and many workers cannot apply in time. Workers who could not access the policy or face difficulty securing approval are left with the choice to remain and work unauthorized or return home.

- The establishment of an expedited and accessible review process that provides workers with full access to reasons for refusals.
- Enactment of an official firewall policy around applications to address workers' concerns that information disclosed during the application or interview could have negative consequences in their future dealings with immigration authorities.

However, even if access to the policy was substantially improved for workers needing to exit abusive employment conditions, the permit does not fundamentally alter the conditions that lead to the workers' vulnerability to abuse and forced labour in the first place—their dependency on employers for legal status in the country.

Furthermore, it is doubtful that the policy will serve as an effective incentive for employers to comply with program conditions and to not abuse or mistreat migrant workers in their employment. Service Canada's failure to systematically track the outcomes of inspections that were triggered by an approved OWP-V application meant the data gathered from the government could not show to what extent the OWP-V policy has increased the likelihood that abusive conduct will be detected and penalized. However, even if every approved OWP-V application led to a finding of non-compliance and the imposition of a penalty, and even if every approved open work permit issued helped identify a new and different abusive employer, the proportion of foreign worker employers affected by the policy would be insignificant. As a matter of fact, the number of OWP-V permits issued in 12 months (IRCC 2020)[9] relative to the annual number of Canadian employers that receive authorizations to hire under one of the tied-worker schemes is minuscule (approximately 0.016%) (Department of Employment and Social Development Canada 2021).[10] In this context, any "threat" for Canadian employers that a migrant worker in their employ may be able to obtain the OWP-V permit that would lead to the imposition of a penalty is ultimately very remote. As such, the limited case-by-case issuance of an open work permit as an exceptional measure that cannot be expected to counteract the profoundly entrenched structural impunity of abusive employers that the employer-specific work permit has established (and workers' precarious immigration status more generally).

In other words, even if the OWP-V policy were to be improved to the point where all the workers who needed the policy could access it and be approved, even without external assistance, it would not alter the initial power imbalance between employers and migrant workers nor reduce their risks of experiencing rights violations while in Canada. It is only an individual remedy that becomes available once abuse has already occurred—it does not prevent or minimize the initial risk of abuse. In this context, extensive and far-reaching reforms that would allow migrant workers to freely circulate in the labour market and thus have a minimal capacity to exercise their rights in the country are necessary to fulfill Canada's commitment to respect the labour rights and the fundamental freedoms of all migrant workers.

**Author Contributions:** Conceptualization, E.D.-P. and H.D.; methodology, E.D.-P. and H.D.; formal analysis, E.D.-P. and H.D.; investigation, E.D.-P. and H.D.; data curation, E.D.-P. and H.D.; writing—original draft preparation, E.D.-P., H.D. and K.B.; writing—review and editing, E.D.-P., H.D. and K.B.; supervision, H.D.; project administration, H.D.; funding acquisition, E.D.-P. and H.D. All authors have read and agreed to the published version of the manuscript.

**Funding:** This research was received partial financial support from the Sisters of St. Joseph of Toronto. This article is published as part of the Special Issue Vulnerability and the Legal Protection of Migrants: A Critical Look at the Canadian Context. The publication of this article is made possible by the VULNER project which is funded by the European Union's Horizon 2020 research and innovation program (grant agreement no. 870845). For more information about this project, visit https://www.vulner.eu/.

---

[9] Between June 2019 and August 2020, 630 applications were approved, a monthly average of 42 open work permits issued to workers as a result of employer abuse (630/15).

[10] During the 'low' year of 2020 (due to pandemic obstacles), 31,424 Canadian employers still managed to be issued authorizations by the federal government to hire under one of the tied-worker schemes.

**Institutional Review Board Statement:** This research was conducted outside of an academic context and did not require an ethics approval. However, informed consent was obtained from all research participants, by providing each participant with a consent form that they were required to real, fill out, sign, and return to the authors. The consent form explained the objectives of the research and provided information about how data would be used, stored, and reported to ensure that the identity of participants would not be revealed. It also informed participants that they could withdraw from the research project at any time and for any reason, or refuse to answer any question.

**Informed Consent Statement:** Informed consent was obtained from all subjects involved in the study.

**Data Availability Statement:** Not applicable.

**Conflicts of Interest:** The authors declare no conflict of interest.

## Appendix A

Organizations that directly support migrant workers and have assisted workers applying for the OWP-V permit were directly contacted. Additionally, a general call for participation was sent out through various migrant worker rights networks to raise awareness about the project. The organizations that agreed to participate selected their representative for the interview. Snowballing sampling was used to identify other groups and individuals who would be interested in participating in this research.

The participants signed a consent form that detailed how their information would be protected, whether the interview could be recorded and transcribed, their right to withdraw, and their right to request anonymity in reporting the data. Most participants selected anonymity. Given the small sample size and the risk of identification through a process of elimination (if some participants were identified and others not), it was decided that all data would be presented anonymously.

The ten semi-structured interviews were conducted over the phone. The open-ended questionnaire was used with some degree of flexibility to allow the interview participants to focus on their areas of expertise. In some cases, a summary of topics was sent beforehand so that the participants could verify certain information before the interview, which was particularly useful with organizations that have more than one person that assists with these types of applications.

The interviews were transcribed and then analyzed to identify the recurring themes. Prevalent themes that emerged included various obstacles to the accessibility of the policy, issues with the processing of applications, limitations regarding the benefits of the policy, and the type and extent of assistance provided to workers trying to access the policy.

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
