# Peer review of "Band-Aid on a Bullet Wound—Canada’s Open Work Permit for Vulnerable Workers Policy"

_laws, 2022_

Round 1

Reviewer 1 Report

Thank you for the opportunity to review this paper. The article contains important findings that expose key harms caused by employer-specific work permits and Canada’s temporary immigration streams more generally. As such, I would like to see this research published. However, I feel the article requires major changes. Most importantly, the article should be revised so as to engage meaningfully with other scholarships and scholarly debates and to clearly outline its contributions to the literature. I also feel the article should engage more explicitly with the fact that this research was done through or in collaboration with a community organization.

The paper presents an insightful analysis of the Open work permit for vulnerable workers. The author highlights important dynamics at play in this permitting process and the TFWP more generally. The article shows that workers face major barriers to accessing the permits and that the program itself is not effective at mitigating abuse or protecting workers. The author makes a compelling case that this type of individualist response is not capable of effectively mitigating harm to migrant workers because such a response does not address the multifaceted power imbalances between employers and migrant workers. I especially appreciate the nuanced attention to a range of barriers that workers face to benefiting from these permits, from a simple lack of access to technology to the complex psychological element of workplace abuse in a working arrangement characterized by intertwined employment and immigration power dynamics. The author thoroughly documents the intersecting problems with the OWP-V and sheds light on the complex ways these problems play out for workers trying to navigate abuse.

But the significance of these findings is limited by the author’s lack of engagement with the other research and scholarly literature. There is no literature review, nor does the author make clear their contribution to the scholarly literature. The reader is left without any clear sense of the previous research done in this area, the bodies of literature that inform the article, or the disciplines, sub-disciplines, or debates to which the author wishes to contribute. At one point the author suggests that something has been “repeatedly flagged by researchers” (p. 16) but does not cite any of the literature to which they are referring. The one reference in the introduction to literature on Canada’s migrant worker programs and policies is an article from 2012. Much has been written on this topic since that time and the author should engage with this body of research. For Canadian-specific research, as a starting point, I would suggest reviewing the work associated with the On the Move project, based at Memorial University. Engagement with scholarly literature will provide the reader with more relevant background information on the topics at hand, acknowledge work that has come before, and frame the contributions of this paper.

I see this engagement with the literature as a basic precondition for publication. But, in the specific case of this article, engagement with the scholarly literature is also important to distinguish this article from the report on which it is based. The author mentions that the research that informs this paper was done through (or perhaps in collaboration with) a community organization. The article under review, they explain, is based on a report written on the basis of that research. But the author does not explain the relationship between the report and the article under review. I feel the author should explain how and why the article makes a unique contribution that differs from the report—presumably by making an explicit engagement with the scholarly literature on this topic.

A related suggestion is that the author should, in general, provide more detail about the character of their working relationship with the community organization and this research project/collaboration. Did the author conduct the research as a service to the organization? Or was this a research collaboration? This strikes me as an interesting and potentially unique component of the research and the author’s methods that should be highlighted.

The two structural weaknesses described above are the most pressing. Some additional more specific suggestions that may be helpful include:

  • Provide more background detail about Canada’s guest worker programs and the emergence of the OWP-V.
  • Reduce detail in the methods section and dedicate this space to a literature review and other engagement with literature. Paragraphs 3, 5, and 6 in the “semi-structured interviews” section, for example, could be eliminated or shortened.
  • Where possible, include some details related to geography, industry, and occupation of the workers being discussed, even if it is general (as a hypothetical example, “most of the interviewees worked with workers in _____ industry in the surrounding rural areas”). Does the author know any details of the industries, jobs, or places of individual workers introduced through the interviews? Were they able to determine whether these factors shaped their experiences with the OWP-V?
  • Related to the above, justify the narrow geography of the study, explain the applicability to other Canadian contexts, or offer some informed speculation about how other contexts might differ. If you are able to offer more detail on industry, geography, etc., readers will be more able to determine the applicability of this study to other contexts.
  • The author makes specific claims that should be supported by secondary sources. For example, the author references the structural vulnerability imposed on workers through employer-tied migrant worker schemes (e.g., p. 2). This vulnerability has been documented in numerous ways and that research should be acknowledged.
  • On p. 3 the author states that one of the paper’s objectives is to assess whether the OWP-V “has been implemented efficiently from a client-experience perspective.” I think this requires some explanation, as it is not obvious to me how one could assess this without interviewing workers directly. I appreciate the good reasons (and the author’s frankness about) why they did not interview workers, but I wonder if a client-experience perspective is possible to glean from the data collected or if this objective could be reframed?
  • I thought that the data collected directly from IRCC was an interesting element of the paper and that this component could be emphasized more. Please also explain how this request worked (on p. 5) in such a way that others would be able to replicate it.
  • Please be sure to explain all acronyms.

Reviewer 2 Report

Thank you for this very interesting paper, I think it can be published with no major revision. To facilitate reading, I would shorten section 2.1 and provide the detailed information about the interviews in an appendix. I would instead expand the last section by discussing more extensively what could be done to address the issues that emerged in the interview. Even if sometimes it is clear what the author(s) think should be done to address those issues by reading the previous subsections, I think the article solid benefit from a broader general discussion at the end. 

Reviewer 3 Report

The article under review is an interesting work examining Canada's current labor policy towards migrants. I have two more fundamental reservations about the article under consideration - suggestions for improvement:

1. supplement the literature review with other sources from the Web of Science and Scopus databases, for example:
- Žofčinová, V., Horváthová, Z., Čajková, A. Selected social policy instruments in relation to tax policy. Social Sciences, 2018, 7 (11), 241
- Peráček, T. Human resources and their remuneration: managerial and legal background. RELIK 2020: Reproduction of human capital - mutual links and connections. 2020, pp. 454-465
- Shevchuk, O., Matyukhina, N., Babaieva, O., Dudnikov, A., Volianska, O. (2021). The human right to security in the implementation of the concept of the "right to health protection". Juridical Tribune, 11 (3), pp. 535-548, doi: 10.24818 / TBJ / 2021/11 / 3.08
- Miežiene, R., Krutuliene, S. (2019: The Impact of Social Transfers on Poverty Reduction in EU Countries. Baltic Journal of European Studies. 9 (1), pp. 157-175
- Charaia, V., Chochia, A., Lashkhi, M. (2021). Promoting fintech financing for SME in S. Caucasian and baltic states, during the COVID-19 global pandemic. Business, Management and Economics Engineering, 19 (2), pp. 358–372
- Srebalová, M. & Vojtech, F. (2021). SME Development in the Visegrad Area. Eurasian Studies in Business and Economics, 17, pp. 269–281, doi: 10.1007 / 978-3-030-65147-3_19
- Funta, R. (2011). Economic law and economic crisis. Where do we go from here? Economic, legal and political dimension. Danube, 2011, 2011 (1), pp. 65–71
Toplak, J., Brezovnik, B. Information delayed is justice denied: Lengthy procedures deny the right to access information. Informatologia, 2019, 52 (1-2), pp. 1-8

2. supplement the conclusion / discussion as well as suggestions of own recommendations on how to eliminate / eliminate the researched problem

Round 2

Reviewer 1 Report

24 March, 2022

I appreciate the care the author has taken to address my concerns. My major concerns have all been addressed. In particular, the article now engages meaningfully with other scholarship and outlines its contributions. The author has also clarified the relationship between this paper and the previously published report as well as the relationship between the community organization and the author of this article. My smaller concerns have also been addressed. Overall, I think these changes have distinguished this paper from the report and strengthened its scholarly framing.

I look forward to seeing this paper published. I think it presents important findings about the harms caused by employer-specific work permits and the imperative of a more systemic solution to addressing these injustices.